# Bioavailability Enhancement Techniques for Poorly Aqueous Soluble Drugs and Therapeutics

**DOI:** 10.3390/biomedicines10092055

**Published:** 2022-08-23

**Authors:** Dixit V. Bhalani, Bhingaradiya Nutan, Avinash Kumar, Arvind K. Singh Chandel

**Affiliations:** 1Academy of Scientific and Innovative Research (AcSIR), Ghaziabad 201002, India; 2Membrane Science and Separation Technology Division, Central Salt and Marine Chemicals Research Institute (CSIR-CSMCRI), G. B. Marg, Bhavnagar 364002, India; 3Department of Biosciences and Bioengineering, Indian Institute of Technology Bombay, Mumbai 400076, India; 4School of Science, STEM College, RMIT University, Melbourne 3000, Australia; 5Center for Disease Biology and Integrative Medicine, Faculty of Medicine, The University of Tokyo, 7-3-1 Hongo Bunkyo-ku, Tokyo 113-8655, Japan

**Keywords:** solubility, bioavailability, dissolution, nanoparticles, encapsulation, BCS classification, low water solubility, solid dispersion, crystal engineering, solid lipid nanoparticles, drug conjugates, micronization, micelles

## Abstract

The low water solubility of pharmacoactive molecules limits their pharmacological potential, but the solubility parameter cannot compromise, and so different approaches are employed to enhance their bioavailability. Pharmaceutically active molecules with low solubility convey a higher risk of failure for drug innovation and development. Pharmacokinetics, pharmacodynamics, and several other parameters, such as drug distribution, protein binding and absorption, are majorly affected by their solubility. Among all pharmaceutical dosage forms, oral dosage forms cover more than 50%, and the drug molecule should be water-soluble. For good therapeutic activity by the drug molecule on the target site, solubility and bioavailability are crucial factors. The pharmaceutical industry’s screening programs identified that around 40% of new chemical entities (NCEs) face various difficulties at the formulation and development stages. These pharmaceuticals demonstrate less solubility and bioavailability. Enhancement of the bioavailability and solubility of drugs is a significant challenge in the area of pharmaceutical formulations. According to the Classification of Biopharmaceutics, Class II and IV drugs (APIs) exhibit poor solubility, lower bioavailability, and less dissolution. Various technologies are discussed in this article to improve the solubility of poorly water-soluble drugs, for example, the complexation of active molecules, the utilization of emulsion formation, micelles, microemulsions, cosolvents, polymeric micelle preparation, particle size reduction technologies, pharmaceutical salts, prodrugs, the solid-state alternation technique, soft gel technology, drug nanocrystals, solid dispersion methods, crystal engineering techniques and nanomorph technology. This review mainly describes several other advanced methodologies for solubility and bioavailability enhancement, such as crystal engineering, micronization, solid dispersions, nano sizing, the use of cyclodextrins, solid lipid nanoparticles, colloidal drug delivery systems and drug conjugates, referring to a number of appropriate research reports.

## 1. Introduction

The solubility of the drug, the solution, and its gastrointestinal permeability are essential factors that control the amount of absorption and absorption speed, along with the bioavailability of the drug [1]. A necessary factor involved in absorption after a drug’s oral administration is the aqueous solubility of therapeutics. The drug solubility is the dissolution rate at which the drug molecule or the dosage form allows for entering the solution, and it is essential when the time of dissolution is restricted [2]. However, the drug’ bioavailability depends on water solubility, dissolution rate, drug permeability, susceptibility to efflux mechanisms, and first-pass metabolism [3]. “Solubility” has been well-defined as the quantity of solute, which dissolves in a quantity of solvent. Quantity refers as the concentration of the solute in a saturated solution at a definite temperature. The solubility has been represented through multiple concentration expressions—for example, parts, percentage, molality, molarity, volume fraction, and mole fraction. Qualitatively, solubility can be termed as a spontaneous collaboration between two substances to create a homogenous dispersion at the molecular level. The solute can be referred to as at equilibrium with the solvent in a saturated solution [4,5,6].

In recent years, according to drug detection, the number of poorly soluble drugs has increased, with 70% of novel medications presenting low aqueous solubility [7]. The controlling factors for the in vivo bioavailability of oral formulations of these drugs are the low solubility and low dissolution rate in gastrointestinal solutions. Thus, an essential issue relating to drug development has been recognized as being in vitro dissolution and increasing the speed of dissolving low solvable drugs in addition to improving their bioavailability, representing a major task for pharmaceutical experts. [8,9].

For the medicinal product to be immersed, it must exist in a water-soluble state at the place of absorption [10,11,12]. The solubility and permeability represent promising factors for in vivo absorption. They can be improved through solubility enhancement techniques [13].

Rebamipide belongs to BCS class IV drugs. It exhibits poor bioavailability and has difficulties in formulation preparation for oral administration. Due to its limitations, it is impossible to formulate a self-nano emulsifying drug delivery system (SNEDDS) formulation using this substance. To enhance the solubility of this drug, a SNEDDS formulation was prepared by complexing rebamipide with its counter ion. Tetra-butyl phosphonium hydroxide (TBPOH) and NaOH were used as counter ions to prepare a complex. Okawa et al. reported that the complexes prepared with rebamipide, Reb–TBPOH complex and Reb–NaOH complex, showed enhanced solubility and absorption in in vitro as well as in in vivo studies. Asper recent scenario of pharmaceutical developing market Figure 1 demonstrates the development of different formulations and their ratio as per BCS classification.

Quercetin is a highly hydrophobic drug, which is a polyphenolic flavonoid. It finds application as an antioxidant. It exhibits anti-proliferative and chemopreventive characteristics. It was found to have potential against colon, lung, ovarian and breast cancer but, due to its poor solubility and low bioavailability, its pharmacological effects are limited. Kakran et al. used bottom-up and top-down approaches as side reduction techniques to prepare nanoparticles [15]. High-pressure homogenization and bead milling approaches were applied top-down and EPN-evaporative precipitation of nanosuspension was applied as a bottom-up technique to prepare quercetin nanoparticles with enhanced solubility and bioavailability.

Ting et al. drew special attention to specialized polymer design strategies and commercial products, which can enhance the solubility of extremely hydrophobic drugs, and can minimize the probability of drug recrystallization. The specially prepared excipients are molecularly customized polymers which can restructure the APIs into potential oral medication [14]. United States Pharmacopeia (USP) has already approved several such specialized polymers as an excipient for the preparation of solid oral formulations. Natural derivatives and some synthetically prepared polymers were frequently used to enhance the bioavailability of amorphous drugs. At present, several amorphous drugs are available on the market, as listed in Table 1. Hydroxypropyl cellulose (HPC), hydroxypropyl methylcellulose (HPMC), polyvinylpyrrolidone), PEG (Polyethylene glycol), HPMCAS (hydroxypropyl methylcellulose acetate succinate (PVP), polyvinylpyrrolidone vinyl acetate (PVP-VA) and several other specialized polymers were approved as excipient by FDA. These were used with drugs such as verapamil (Product trade name: ISOPTIN-SRE, excipient: HPC/HPMC by Abbott Laboratories), Nilvadipine (product trade name: Nivadil, excipient: HPMC by Fujisawa Pharmaceuticals, Tokyo, Japan ), tacrolimus (product trade name: PROGRAF, excipient: HPMC by Astellas Pharma US Inc., Northbrook, IL, USA), nabilone (product trade name: Cesamet, excipient: PVP by Valeant Pharmaceuticals, Bridgewater, NJ, USA ), griseofulvin (product trade name: GRIS-PEG, excipient: PEG by Pedinol Pharmacal Inc. & Novartis, Bridgewater, NJ, USA ), ritonavir (product trade name: NORVIR, excipient: PVA-VA by AbbVie Inc., Chicago, IL, USA) and telaprevir (product trade name: INCIVEK, excipient: HPMCAS by Vertex Pharmaceuticals, Boston, MA, USA) [2,16,17,18]

Johnson et al. developed seventeen types of poly(*N*-isopropyl acrylamide) (PNIPAM)-based excipients differing in molar mass and with a variety of end groups. These polymers were studied for their capability to enhance the water solubility of phenytoin. To improve the solubility of phenytoin, Johnson et al. synthesized three polymeric excipients: PNIPAM, poly(*N*,*N*-dimethyl acrylamide) (PDMAm) and PHEAmpoly(*N*-hydroxyethyl acrylamide) (PHEAm). Synthesized polymers were investigated for their solubility enhancement effect on phenytoin. Johnson et al. described a detailed understanding of the critical importance of PNIPAM for the solubility enhancement of phenytoin [20,21].

Chen et al. worked on the drug solubility enhancement of docetaxel (DTX). Docetaxel finds application as a chemotherapeutic agent for cancer treatment. Chen prepared three types of inclusion complex among docetaxel and H1-3 (ethylenediamine modified beta cyclodextrins) with ethylene, propylene and butylene parts. Chen proved that the complexation of DTX with H1-3 is an effective tactic to enhance the solubility and prepare a less toxic and highly active DTX formulation. This approach can maximize its clinical applicability in cancer treatment [22].

In recent decades, solid dispersion technology was extensively studied to develop an amorphous carrier to increase the bioavailability, solubility and dissolution rate of drugs with poor water solubility. The solid dispersion preparation methodology and selection of appropriate carriers will play a critical role in its biological behavior [23]. Here are some techniques to prepare amorphous solid dispersions to enhance the bioavailability, solubility and therapeutic efficacy of drugs: (a) cryogenic processing techniques, (b) freeze drying, (c) fluid-bed coating, (d) spray drying, (e) microwave irradiation, (f) co-precipitation method, (g) electrostatic spinning, (h) supercritical anti-solvent (SAS), (i) HME technique, (j) Meltrex^TM^, (k) melt agglomeration, and the (l) KinetiSolVR dispersing (KSD) technique.

The drug solubility is mainly influenced by its chemical arrangement and the conditions of its solution. The molecular assembly defines its molecular volume, crystal energy, hydrogen bonding, ionizability, and lipophilicity, which determines the drug solubility. pH, additives, time, temperature, cosolvents, and ionic strength will affect the solution conditions. Pharmaceutical compounds with poor water solubility can intensely decrease output in drug discovery and development.

A “good compound” must reach the target site’s inactive focuses. Solubility of compound affects their absorption, permeability, and potency. Highly potent and permeable compounds are suitable for low aqueous solubility.

### 1.1. Biopharmaceutical Classification System

BCS is one of the most applied scientific classification systems of drug substances based on their permeability and solubility as described in Figure 2. There are two significant factors which regulate the speed and scope of oral drug absorption that is aqueous solubility and intestinal permeability [24].

The compounds of drugs can be categorized into four classes, according to BCS.

When the maximum potency of the drug is soluble in ≤250 mL of the aqueous standard, it means drugs are highly soluble, and their range of pH is 1.0 to 7.5; otherwise, the drug elements are measured as not very soluble. Biopharmaceutical researchers are continuously trying to make a biologically mimicking system that can match conditions such as gut pH, food content, and peristalsis to predict in vivo performance precisely. Several biopharmaceutical research developments were made between 1960 and 1970; numerous studies were carried out and established the relationship between the effect of dissolution and formulation parameters on drug bioavailability. For the proper evaluation of any formulation’s dissolution rate, the first dissolution test apparatus was introduced in 1970, USP apparatus I (basket type), and afterwards, another USP Apparatus II (paddle type) was introduced [25]. Using this apparatus, it has been possible to predict the in vivo performance of the formulation from the in vitro tests. However, because the in vivo performance of every formulation depends on several variables, improvements have been made to the in vivo performance of dosage forms. Figure 3 provides a chronological list of some of the significant studies conducted in this area.

### 1.2. Importance of Solubility Enhancement

The main difficulty in the advancement of a new chemical entity is low aqueous solubility. For oral drugs, the most rate-limiting factor is solubility and reaching its concentration aspirated in full circulation for the pharmacological response. To obtain an approved concentration providing the necessary pharmacological reaction, solubility is one of the essential parameters [27]. Hydrophobic drugs usually require high doses and need high dosage regimens to influence therapeutic plasma concentrations after administration [28].

The oral administration of drugs is the most accessible and generally applied way of delivering drugs thanks to flexible administration, profitability, high compliance by the patient, flexibility in the dosage design, and fewer sterility restrictions. For this reason, most generics companies prefer to yield bioequivalent oral pharmaceuticals [3]. However, the main problem with oral administration is related to its dosage form design and its low bioavailability.

The factors affecting oral bioavailability include drug permeability, first-pass metabolism, dissolution rate, aqueous solubility and susceptibility to efflux mechanisms. The most common reasons for low oral bioavailability are associated with low permeability and poor solubility.

In the case of other dosage forms such as parenteral formulations, solubility plays a critical role [29]. Drugs with poor water solubility frequently require a high dosage of the drug to extend therapeutic plasma concentrations following oral administration. The essential criteria include being in an aqueous medium at the absorption site for the absorption of any pharmacological compound. H_2_O is the preferred solvent for preparing liquefied pharmaceutical preparations. A drug’s slow solubility in aqueous media is the foremost difficulty for formulation scientists [30]. Most drugs are weakly basic or weakly acidic, having less water solubility. Almost 40% of NCEs (new chemical entities) are insoluble in water, as discovered in the pharmaceutical sector.

Drug solubility enhancement is the most challenging factor in the field of drug discovery. In the literature, several strategies exist and have been described for the solubility improvement of poorly water-soluble drugs. These techniques are preferred based on specific features such as the characteristics of the drug under consideration, intended dosage form types, and chosen excipient characteristics.

The low dissolution rate and low solubility in the aqueous gastrointestinal fluids lead to inadequate bioavailability. Mainly aimed at group II compounds conferring to the BCS, bioavailability can be improved through enhancing the dissolution rate and solubility of the drug in gastrointestinal fluids. The ratio restrictive factor for the class II BCS is the release of the drug from the dosage form and gastric fluid solubility and not the absorption, and thus enhancing the solubility will improve the bioavailability of BCS class II drug molecules [3,30,31].

The negative impact of drugs having poor solubility involves less bioavailability and poor absorption for IV dosing, due to that high cost (frequent high-dose administration) are challenges [29].

## 2. Techniques for Solubility Enhancement

In Table 2, different approaches for solubility enhancement are explained with their advantages and limitations. The major techniques have listed in Table 3.

### 2.1. Physical Modifications

#### 2.1.1. Particle Size Reduction

The drug solubility depends on its particle size. Large particles provide a low surface area, which results in less interaction of particles with the solvent. One of the methods to increase the drug’s surface area is to reduce its particle size, which improves its dissolution property.

**Micronization:** The process of producing drug particles in micron size by using the physical method. The methods widely used for increasing BCS class II drugs’ solubility are freeze-drying, crystallization, spray drying, and milling [58].

Size reduction in the conventional time is achieved through mechanical methods, i.e., grinding, milling, and crushing of heavier particles to reduce their size by applying friction, pressure, attrition, shearing or impact. For mechanical micronization, ball mills, jet mills and high-pressure homogenizers are utilized. Dry milling is the most preferred micronization method [59].

Micronization raises the dissolution speed rather than the drug’s equilibrium solubility. In various studies, it has been reported these that methods for the reduction in size are used to increase the dissolution and bioavailability through decreasing dimension and increasing the surface area of poorly aqueously soluble drugs.

**Nanosuspension:** Nanosuspension is well-defined as a colloidal dispersion of sub-micron drug elements, stabilized by using a surfactant. To produce a nanosuspension, wet milling and homogenization are used. Milling defragments the active compound in the presence of a surfactant.
Advantages of nanosuspension [60]Enhancement of drug solubility and its bioavailabilityHigher drug loadingSuitable for hydrophobic drugsPassive drug targetingReduction in dosageIncrease in drug’s physical and chemical stability.

Methods for the Preparation of a Nanosuspension: A nanosuspension is primed via two main methods—“bottom-up” and “top-down” technology [61].

Bottom-up technology—This is an assembling technique for the formation of nanoparticles, such as precipitation, melt emulsification, and microemulsion.

Top-down technology—Includes the decomposition of heavier particles into small particles, such as the high-pressure homogenization method and the grinding techniques [62].

Precipitation method—The precipitation technique is employed to produce submicron particles of drugs with poor solubility. Here, drug molecules are solubilized in a solvent, and then that solution is poured into the other solvent in which drug molecules are unsolvable with the occurrence of a surfactant. Quickly mixing the solution with a solvent leads to quick supersaturation of the drug in the solution, which leads to the development of an ultrafine crystalline or amorphous drug. This technique includes two steps: the formation of nuclei and the growth of the crystal. These mainly depend on the temperature. Fundamental necessities for the development of stable suspension are a high nucleation rate and low crystal growth [48,63].

High-pressure homogenization—The high-pressure homogenization method comprises three steps: (i) the dispersion of drug powders in a stabilizing solution to make a presuspension; (ii) the homogenization of the presuspension by the low-pressure, high-pressure homogenizer, sometimes for the previous preparation; and (iii) 10 to 25 cycles of final homogenization at a high pressure until the nanosuspensions attain the desired size [64].

Homogenization in aqueous media (Dissocubes) [65].

Homogenization in nonaqueous media (Nanopure).

##### Milling Techniques

(a)Media milling

Media milling includes the formation of drug nanoparticles by impaction within drugs and milling media; impaction provides enough energy for particle breakdown. This technique involves a grinding chamber, which is filled with the drug, a stabilizer, grinding means, and water or some other suitable buffer. The grinders rotate at a very high cutting speed to create the suspension. The major drawback of this method is the residues left behind, which are present in the finished product [40].

(b) Dry grinding

In the early days, nanosuspensions were developed by the wet grinding technique via a pearl ball mill. Now, they are developed via dry milling. In this technique, nanosuspensions are developed using the dry grinding of polymers and soluble copolymers with a poorly soluble drug, after having dispersed it in the liquid medium [66].

##### Lipid Emulsion/Microemulsion Template

The nanosuspensions can be prepared simply by the dilution of the emulsion, which is molded through a partly miscible solvent with water according to a dispersed phase. This method is suitable for drugs soluble in volatile organic solvents or partially miscible in water. In addition to that, microemulsion templates can also be used to create nanosuspensions. Physically, a microemulsion comprises a dispersion of oil and water and stabilizer (cosurfactant or surfactant) fluids, which are immiscible. The drug molecules are introduced in the internal phase, consist of a microemulsion, or are saturated through drugs of an informal mixture. Griseofulvin nanosuspensions are developed by the microemulsion process using H_2_O, taurodeoxycholate sodium salt, butyl lactate, and lecithin [67].

##### Microprecipitation—High-Pressure Homogenization (Nano Edge)

This technique is a unification of the high-pressure homogenization and microprecipitation technique. This process involves the precipitation of breakable constituents that succeed through fragmentation [48].

##### Nanojet Technology

This technique is termed the reverse flow technique. Nanojet uses a compartment in which a suspension current is distributed into two or more two parts. Both streams exist as high-pressure colloids. Because of the collision, the high shear force generated throughout the procedure shows particle size reduction. The nano-suspensions of atovaquone are developed using the micro fluidification technique. The main limitation of the method is a high amount of cycles over the microfluidizer, and therefore the resulting output possesses a comparatively greater number of microparticles [68]. The nanoparticle techniques can be useful as a screening protocol for studying the safety and preclinical efficacy of NCEs. Nanoparticle-based drug delivery techniques can be helpful in the preparation of existing NCEs with a higher bioavailability and minimum toxicity. In Table 4, a list of several marketed products using nanotechnology was represented [69].

Various approaches based on nanotechnology for the enhancement of oral bioavailability of poorly water-soluble drugs are presented in Table 5.

#### 2.1.2. Modification of the Crystal Habit

***Crystal engineering:*** Crystal engineering is the study of the design, modelling, synthesis, and application of crystalline solids containing a predefined and preferred combination of molecules and ions. Crystal engineering is the exploitation of noncovalent interactions among ionic or molecular components for the rational design of solid-state structures, which show exciting optical, magnetic, and electrical properties [70].

***Hydrates/solvates:*** Solvates are molecule adducts that incorporate solvent molecules in their crystal lattice. When the solvent is H_2_O, then the solvate is called a hydrate [71].

***Polymorph:*** Polymorphs are defined as the phenomenon in which a compound has a different crystal structure but similar chemical composition; hence, due to their different network structures/molecular conformations, they have different physicochemical properties. Polymorphism is a common phenomenon by which many drugs can crystallize into dissimilar polymorphic structures to increase solubility.

#### 2.1.3. Drug Dispersion in Carriers

##### Eutectic Mixtures

When two or more compounds are mixed, generally they do not show phase compatibility with each other to generate a new entity, but, at specific fractions, they prevent the crystallization process of one another, which results in a system having a lower melting point than either of the two starting components [72].

##### Solid Dispersion

When using a hydrophilic matrix and a hydrophobic drug, these two different components become molecularly dispersed in amorphous particles (clusters) or crystalline particles [73,74,75,76]. Solid dispersion techniques are discussed in Table 6, along with their BCS class—for example, drug molecules and trade name, type of formulation and their therapeutic uses.

***Solvent evaporation method:*** An organic solvent is evaporated after the entire dissolution of both the drug and carrier in solvent evaporation. The dense form is ground, sieved, and dried—e.g., furosemide with eudragits.

***Hot-melt extrusion method:*** In this method, carriers and active pharmaceutical ingredients are prepared by hot-stage extrusion using a co-rotating twin-screw extruder. The dispersion of drug concentration is 40% (*w*/*w*). It is employed for formulating diverse dosage forms—e.g., sustained-release pellets.

***Kneading technique:*** In this technique, the drug carriers with H_2_O are converted into a paste, and then the drug compound is mixed into a paste and pressed for a fixed period; after that, the pressed mixture is passed through the sieve after it gets dried.

***Co-precipitation method:*** A certain quantity of the drug is poured into the carrier solution under continuous magnetic stirring, and the solution must be kept away from sunlight in the co-precipitation method. Then, precipitates are parted via purification through a vacuum, and they should be kept for drying at room temperature to prevent water loss due to the inclusion of complex structures [77].

***Melting method:*** In this method, a mortar and pestle are employed for mixing drugs and their carriers. After mixing, the mixture is heated up to the melting temperature of all of the ingredients to achieve a homogenous dispersion; then it is cooled to obtain a congealed mass. The mass is further crushed and sieved—e.g., albendazole and urea [78].

***Co-grinding method:*** The combination of the carrier and drug is prepared by a blender (blending for a particular fixed time and speed). Then, the prepared mixture is transferred to the vibration ball mill compartment, in which steel balls are further added. Pulverization is carried out, and the sample is taken out and stored at room temperature—e.g., chlordiazepoxide and mannitol [79].

***Gel entrapment technique:*** To prepare a clear and transparent gel, an organic solvent is used for dissolving hydroxyl propyl methylcellulose, and then the drug compound is liquefied in the gel by the application of sonication for a certain time. After the organic solvent is detached from the vacuum, the prepared solid dispersions are size reduced by a mortar and pestle and then separated using a sieve [80].

***Spray-drying method:*** The essential quantity of a drug solubilized in an appropriate solvent and carrier in water (aqueous media). Sonication or some other appropriate techniques are combined and employed for the preparation of a clear solution, which is further spray-dried in a spray dryer [81].

***Lyophilization technique:*** This process was projected as a substitute for the solvent evaporation technique. It is a type of molecular mixing process in which drug compounds and their carriers are combined, solubilized in a universal solvent, frozen and sublimed to obtain a lyophilized molecular dispersion that involves the transfer of mass and heat from the product under preparation [82].

***Melt agglomeration process:*** This process is different from other techniques because the binder itself acts as a carrier to prepare solid dispersion. Apart from this, solid dispersions are created by heating the drug compound, binder, drug and their excipient at or above the melting temperature of the binder or by using a high shear mixer. The drug compound is dispersed in the molten binder by spraying it onto the heated excipient [83]. Due to the easy temperature control and high content of the binder, a rotary processor can be combined as a piece of alternative equipment [84].

##### Solid Solutions

When two components are mixed, they crystallize together in a single homogeneous phase, considered a solid solution.

They are of two types: substitutional solid solutions (random and ordered), and interstitial solutions.

***Substitutional:*** The condition where solute atoms occupy some space in the regular lattice sites of the parent metal (solvent)—e.g., random (Cu-Ni) or ordered (Cu-Au).

***Interstitial:*** The condition where solute atoms occupy the space in interstitial positions—e.g., (steel C solute atoms in Fe).

Solid solutions can generally attain a quicker dissolution rate than the corresponding eutectic mixture

#### 2.1.4. Solubilization by Surfactants

##### Microemulsion

Microemulsions are clear, transparent, unstable mixtures of two immiscible liquids—for example, water and oil. Emulsions are alleviated through an interfacial film formed by surfactants [85].

##### **Components** **of** **microemulsion**

***Aqueous phase:*** Water is used as the most common aqueous phase. Because of the phase behavior, the pH of the aqueous phase needs to be adjusted. In the case of microemulsions used for parenteral administration, the aqueous phase should be isosmotic to the blood, which is maintained by sodium chloride, glycerol, dextrose, and sorbitol.

***Oil phase:*** Built on the method of drug administration and its nature, the selection of oil is performed. The oil should exhibit good solubilization potential for that drug. The oil changes shape and can inflate the surfactant tail assembly. The unsaturated and saturated fatty acids experience an intensification of their penetration. These fatty acids will increase permeability by interrupting the dense lipids and filling the extracellular spaces of the stratum corneum. For the enhancement of skin penetration, oleic acid is employed in unsaturated fatty acids, and the penetration effect is different for each drug. Isopropyl palmitate is a well-known fatty acid ester used for permeability enhancement [86].

Recently, the selective tendency is towards the use of semi-synthetic oils, which are highly stable, compared to their natural equivalents. Drugs that have low water solubility must have solubility in the dispersed oil phase to produce an effective oil/water microemulsion arrangement. The size of the drops in the microemulsion will increase if the content of oil increases [87].

***Surfactant:*** Surfactants are compounds with a hydrophilic head and a hydrophobic tail. They are present at the interface of systems, and they alter interfacial tension. They are present in deficient concentration. The main aim of the surfactant is to minimize the interfacial tension between the two systems to a negligible value, which promotes the dispersion process during the formation of a microemulsion. It presents the microemulsion with an appropriate lipophilic character to fit the proper shape. A surfactant can unite both nonpolar and polar groups in a single molecule. The selection of a surfactant molecule is performed based on hydrophilic lipophilic balance (HLB) value. This HLB value suggests the kind of emulsion formation (whether it is an o/w or w/o emulsion) [88].

***Co-surfactants:*** Co-surfactants are amphiphilic, accumulate at the interfacial layer, and increase the fluidity of interfacial film by penetrating the surfactant layer. Single-chain surfactants are unable to decrease the interfacial tension of o/w to form a microemulsion. Chain alcohols are used, utilizing co-surfactants to increase the fluidity of the interface. Ethanol and 1-butanol (medium-chain) are used as permeation enhancers. The relationship between surfactant and co-surfactant is the main factor to consider.

##### **Classification** **of** **microemulsion**

Depending on the composition, microemulsions are classified into three kinds:Oil-in-water microemulsions (o/w)Water-in-oil microemulsions (w/o)Bi-continuous microemulsions

The interface of the components in the three microemulsions must be stabilized with an adequate mixture of surfactants and/or surfactants as a stabilizing agent.

Methods for the preparation of a microemulsion [89,90,91]:(i)Phase titration method(ii)Phase inversion method.

##### Self-Emulsifying Drug Delivery Systems (SEDDS)

This scheme is employed for solving low bioavailability problems of poorly soluble and extremely porous drug molecules. Hydrophobic drug molecules can be liquefied in this system. When the constituents of the SEDDS are delivered in the lumen of the gastrointestinal tract, they come into contact with the gastrointestinal fluid, which leads to the development of a fine micro/nanoemulsion, due to which this mixture is termed as a self-emulsification in situ emulsion. This additionally leads to drug solubility, which is consequently engaged through lymphatic pathways, bypassing the hepatic first-pass effect. The bioavailability-improving property has been linked to several properties in vivo of the lipid preparations [92].

Processes for self-emulsification:Self-nano emulsifying drug delivery system (SNEDDS)Self-micro emulsifying drug delivery system (SMEDDS)

Composition of a self-emulsifying drug delivery system:

***Active Pharmaceutical Ingredients (APIs):*** Generally, a self-emulsifying drug delivery system is employed for the solubility enhancement of drugs with low aqueous solubility; drugs of BCS class II are mostly preferred—e.g., itraconazole, naproxen, vitamin E, mefanimic acid, danazol, nifedipine, simvastatin etc.


**
*Excipients used in SEDDS:*
**
OilsSurfactantsCo-surfactantsViscosity enhancersPolymersAntioxidant agents


Table 7 represents the details of various marketed parenteral microemulsion products [69].

#### 2.1.5. Complexation

The complex shares a connection between two or more molecules, which develop an entity, unrelated to a definite balancing. This depends on comparatively weak forces, i.e., hydrogen bonds, hydrophobic interactions, and London forces [93]

##### Stanching Complexation

Stanching complexes are generally produced as overlapping planar domains of aromatic compounds, and the nonpolar groups lead to the removal of H_2_O through strong hydrogen bonding connections. These are particular particles which are known to produce stanching complexes, i.e., anthracene, benzoic acid, pyrene, salicylic acid, methylene blue, nicotinamide, ferulic acid, theobromine, gentisic acid, naphthalene, purine, and caffeine.

##### Inclusion Complexation

Inclusion complexation has been developed using an infusion of a nonpolar particle or a section of the guest particle added to the cavity of different particles or an assembly of molecules (which is termed the host). The main physical prerequisite for complexing the inclusion is a perfect adaptation for guest molecules in the host cavity. The cavity of the host particle should have enough space for accommodating the guest molecule, and it should be small enough that it can eradicate H_2_O as the association between H_2_O and the nonpolar domains of the host molecule and the guest molecule decreases. α, β and γ-cyclodextrin are the three types of naturally occurring CDs. Cyclodextrin is used in complexation for solubility enhancement. The embodiment of cyclodextrin is a molecular phenomenon. In the case of cyclodextrin, one guest particle can combine through a cavity and create a stable association. In the cyclodextrin molecule, a shallow external activity is hydrophilic, and an internal activity is hydrophobic; this is because of the organization of the hydroxyl group inside a particle. The inclusion complexation of cyclodextrin involves the examination of positions for either a one-step reaction or a sequential two-step reaction, in which there is the involvement of structural transformation. Cyclodextrins increase the water solubility of drug molecules by inclusion complexation. The complex of cyclodextrin with clofibrate, rofecoxib, melarsoprol, celecoxib, cyclosporin A, taxol, etc., will improve the solubility of the drug [74,94].

Manufacturing techniques for complexation/inclusion complexation:Kneading methodMicrowave irradiation methodCo-precipitate methodLyophilization/freeze-drying techniqueSpray drying

##### Peptide Complexation

The use of protein nanoparticles in the delivery of materials such as genetic materials, poorly water-soluble drugs, peptide hormones, growth factors, DNA, and RNA has many advantages. The advantages of protein nanoparticles over other colloidal carriers include their stability and ease of manufacture. As a result of an easy, cost-effective, and eco-friendly synthesis process, proteins from various sources can be synthesized into nanoparticles, which require fewer chemicals than nanoparticles from other materials and are expected to have a high potential for in vivo use [95]. Moreover, bioactive substances and drugs will absorb more efficiently and be more bioavailable when their water solubility is increased by complexation. This will allow the drug to be administered at a lower dose to achieve the desired effect, and it will reduce the chances of side effects. As a result of the formation of the bioconjugate, the bioactive substances will also be more resistant to digestion. Through this technique, medicine for human health will be improved.

Chang et al. reported that the improvement of the hydrophobic encapsulation of curcumin in egg white protein nanoparticles and the investigated compound’s encapsulation in egg white protein nanoparticles effectively reduced the degradation ratio as well as protecting the antioxidant activity of encapsulated curcumin [96]. In another study, Prathap et al. reported that a nanoassembly of lysozyme-conjugated curcumin produced a nanoconjugate with enhanced antibacterial, antioxidant, and anticancer activity [97]. The water solubility of a poorly water-soluble nonsteroidal anti-inflammatory drug, indomethacin, was enhanced through complexation with the casein hydrolysate (Figure 4). Similarly, the peptide complexation can enhance the solubility of poorly water-soluble therapeutics [98].

#### 2.1.6. Cryogenic Techniques

Cryogenic techniques are employed to improve the dissolution speed of drugs by formulating an amorphous drug of the nanostructure with a high degree of porosity at minimal temperatures. Afterwards, on completion of cryogenic treatment, the powder is dried via the drying method (vacuum, spray, and lyophilization) [99,100,101]. Various cryogenic methods are mentioned in Figure 5.

### 2.2. Chemical Modifications

#### 2.2.1. pH Adjustment

This plays a critical role in drug solubility. It can influence the aqueous solubility of drugs. By varying the solution pH, one can alter the charge state of the drug molecules. If the pH of the solution is such that a particular molecule carries no net electric charge, the solute often has minimal solubility and precipitates out of the solution. The pH at which the net charge is neutral is called the isoelectric point (sometimes abbreviated to IEP) [102].

#### 2.2.2. Hydrotrophy

This is a solubility sensation; using it, the water solubility of the solute can be enhanced by the excess addition of a second solute. The term hydrotrophy was used in earlier reports to describe non-micelle-forming materials, either solids or liquids, organic or inorganic, which are proficient in improving solubility of insoluble substances [103].

#### 2.2.3. Co-Crystallization

Co-crystals are the complexes of non-ionic supramolecular materials. They can be utilized to address issues regarding physical properties, i.e., drug solubility, bioavailability, and stability, without affecting the chemical structure of APIs. Co-crystals are prepared using two or more different molecular units, in which the weak forces are intermolecular interactions such as π–π stacking and hydrogen bond interactions. The composition and molecular interaction of pharmaceutical compounds will be changed by co-crystallization, and it is accepted as a good option to optimize the drug characteristics. Co-crystals will offer various routes, where any APIs can be crystallized regardless of belonging to acidic, basic or ionizable groups. This can be helpful for compounds with low pharmaceutical profiles due to their nonionizable functional groups [104].

#### 2.2.4. Co-Solvency

When the structural complexity of newly developed entities rises, the H_2_O solubility of the drug decreases drastically. When the water solubility of a compound is much lower than its therapeutic dose, a blend of solvents is employed to obtain high solubility. Co-solvents are used to enhance the drug’s solubility, providing multiple nonpolar groups, thus increasing its aqueous (water) solubility. Co-solvents are necessary for the pharmaceutical formulation, where, sometimes, it may be required to enhance drug solubility [105].

#### 2.2.5. Salt Formation

Acidic and basic drugs have low solubility in water as compared to their salts. For the development of parenteral administration, the most favored strategy is solubility enhancement by salt formation.

#### 2.2.6. Nanotechnology in Pharmaceuticals

Nanotechnology is applicable to promote the solubility of drugs with poor aqueous solubility. Nanotechnology involves extensive investigation and usage of structures and materials at the level of nanoscale, which is up to 100 nm [106]. Micronization is not enough for several NCEs for the enhancement of solubility and oral bioavailability because the micronized material tends to agglomerate, which results in a decline in the effective surface area for dissolution.

Nanotechnology for nanonization:Nanomorphs [107]Drug nanocrystal [108]

### 2.3. Miscellaneous Methods

#### 2.3.1. Supercritical Fluid Technology

Supercritical fluid technology was employed for the first time industrially in the early 1980s in the pharmaceutical sector. During that period, SCF technology was used in pharmaceutical industries for developing pharmaceutical materials through crystallization and precipitation. The SCF. method is safe, eco-friendly, and cost-effective. The low operational parameters (pressure and temperature) make SCFs attractive for pharma research. An SCF survives as a single phase above its critical pressure (Pc) and temperature (Tc) [109,110].

#### 2.3.2. Micellar Solubilization

Micellar solubilization is a technique in which the component is incorporated into or onto the micelles (the component that undergoes solubilization). The most significant characteristics of micelles are their capability to enhance the solubility of the less soluble compounds in water. In this circumstance, solubilization can be described as the natural dissolution of a compound by reversible interaction with micelles of a surfactant in water to develop a thermodynamically stable isotropic solution with reduced thermodynamic activity of the solubilized substance [111]. If the solubility of a compound with poor water solubility is plotted as a function of the surfactant concentration, generally, it is concluded that the drug solubility is much less until the concentration of surfactant touches the CMC. After the surfactant reaches concentrations beyond the CMC., the solubility rises linearly with the surfactant concentration, intimating that solubilization is relevant to micellization. Gliclazide, glipizide, pioglitazone, glyburide, repaglinide, rosiglitazone, and glimepiride are some of the poorly water-soluble compounds which utilize the micellar solubilization technique for improvement in solubility and bioavailability [112].

Other miscellaneous methods are as follows:Direct capsule fillingElectrospinning methodDropping method solution

##### Cyclodextrins

Cyclodextrins are cyclic oligosaccharides with a hydrophilic external surface and a moderately hydrophobic interior cavity. Cyclodextrins create water-soluble inclusion complexes using several hydrophobic drugs with low aqueous solubility [93,113]. Cyclodextrin host molecules are also recognized as have a significant role in the formation of non-inclusion complexes [114]. Extensive studies have been performed on cyclodextrins and their complexes in the last 2–3 decades. These studies have provided vast data on the physical necessities involved in complex development and the associated forces [115]. Hydrophobic drugs, along with cyclodextrin–drug complexes, are identified for aggregate formation in the aqueous medium. Surface-active preservatives and water-soluble polymers are well-known excipients for the solubility of the drug in an aqueous medium [116,117].

There are several methods for complexation with cyclodextrins, such as spray drying, physical mixing, co-evaporation, melting, freeze-drying, and kneading. By means of these techniques, the drug solubility, dissolution rate, and the bioavailability of poorly water-soluble drugs can be upgraded.

##### Solid-Lipid Nanoparticles

Solid-lipid nanoparticles are used for organized and specific targeting in drug delivery. They are biocompatible and biodegradable and have an average particle size ranging from 50 nm and 1000 nm. They comprise a solid hydrophobic phospholipid coating. This coating consists of a lipid matrix that must be at room temperature in solid form, which is dispersed in H_2_O or an aqueous surfactant solution. Solid cores containing the drug are dispersed in the lipid matrix. They are likely to transport both hydrophobic and hydrophilic drugs. [118,119,120,121,122,123,124]. Table 8 represents various examples of drugs developed by using SLN technology [69]. Table 9 lists some examples of the frequently used lipid excipients in lipid-based nanocarriers.

##### Polymeric Micellar Carriers

Incorporating poorly water-soluble molecules in surface-active agents can increase drug solubility and prevent drug precipitation following exposure to the GI environment. Monomeric surfactants, micellar aggregates, and surfactants adsorbed as a film at the interface are the three systems in a surfactant solution where micellar systems occur in dynamic equilibrium. Surfactant concentrations above the critical micellar concentration (CMC) cause the formation of micellar carriers. When dissolved in an aqueous environment, amphiphilic copolymers made up of hydrophobic and hydrophilic building components form micelles [76,187,188,189]. The core and outer shell of the micelles are formed by the hydrophobic domains and hydrophilic tails of the copolymers, respectively. The hydrophobic drug’s contact with the aqueous medium is stabilized by the corona, whilst the lipophilic core acts as a container for loading lipophilic medicines. Lipophilic drugs can have their solubility increased by using micellar carriers incorporated into the micellar core [190]. Table 10 lists the examples of nanocarriers, mainly those utilized for the oral delivery of drugs and therapeutics. Recently, amphiphilic block copolymers were developed as a better alternative for the delivery of hydrophobic drugs and therapeutic molecules with enhanced bioavailability. In the amphiphilic block copolymer, the combination of two or more two types (hydrophilic and hydrophobic) of polymeric segments combines and forms amphiphilic copolymers [190,191,192]; moreover, these amphiphilic copolymers can be synthesized precisely by different polymerization techniques such as ATRP [193], RAFT, and radical polymerization [194]. In these block copolymers [195], the combination and ratio of the different co-polymeric segments are very important, deciding the efficiency of delivery and bioavailability. In Table 11, the important patents of formulations related to nanoparticles for oral drug delivery are listed.

## 3. Conclusions

This review provides a critical summary of previously reported and some currently developing technologies such as formulation design, solid particle techniques, prodrug strategies, crystal engineering, micronization, solid dispersions, particle size reduction technologies, nanosizing, cyclodextrins, solid lipid nanoparticles, drug conjugates, colloidal drug delivery systems, complexation of drugs, usage of formation of emulsion, micelles, microemulsions, cosolvents, polymeric micelles, pharmaceutical salts, prodrugs, solid state alternation, soft gel technology, drug nanocrystals, and nanomorph technology, with a few appropriate research reports and recent advancements. The solubility improvement technique of poorly water-soluble drugs plays a vital part in the formulation development to fulfil the therapeutic action and drug bioavailability of the pharmaceutically active ingredient (drug) at the target site. The pharmaceutical industry screening programs identified that around 40% of new chemical entities (NCEs) face various difficulties at the formulation and development stage. This is mainly due to poor water solubility and less bioavailability. The bioavailability and drug solubility enhancement are significant challenges in the area of pharmaceutical formulations. According to the biopharmaceutical grouping structure, Class II and IV drugs (APIs) have low water solubility, less bioavailability, and poor dissolution. Solving the above-mentioned problems here is based on molecular properties, but researchers view solid dispersions and lipid delivery as the most highly sought-after techniques to solve these for reasonable compounds. The solid dispersions can be in the form of a spray drawing or hot melt extrusion, and lipid delivery can use lipids for either delivering solubilization or improving permeability, and this is applicable in a variety of dosage forms, including soft shells or liquid-filled hard shells as some examples of lipid delivery. So, there are several trends in solubility enhancement in terms of new processes, including new excipients to help poorly soluble molecules; in particular, one of the areas that is being studied extensively is modelling and understanding the poorly soluble molecules and therefore which formulations will work best. Therefore, the ideal solution would be a technique or a polymer formulation that will overcome 100% of the reported solid molecules. This does not exist, but we are progressing towards being able to model molecules, looking at the properties of the molecule to predict which polymer will work best to perform the technique quickly.

The graphical abstract of this article is created with Biorender.com (accessed on 7 August 2022).

## Figures and Tables

**Figure 1 biomedicines-10-02055-f001:**
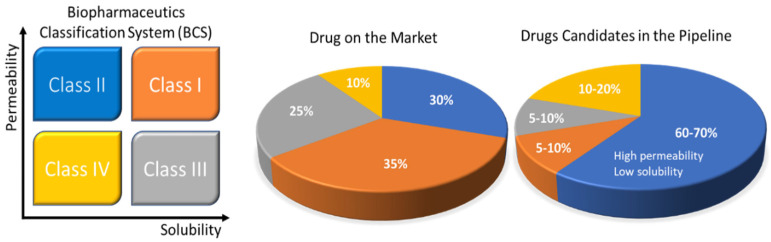
Solubility challenges that plague the oral drug-delivery frontier. The Biopharmaceutics Classification System (BCS) uses drug permeability and solubility as metrics for oral absorption. The four categories include BCS Class I (orange: high solubility, high permeability), Class II (blue: low solubility, high permeability), Class III (black: high solubility, low permeability), and Class IV (yellow: low solubility, low permeability). The pie charts to the right show the estimated distribution of marketed and pipeline drugs by BCS classes. The pharmaceutical company data are also presented in the pie charts. Reprinted and adapted with permission from ref. [14]. Copyright 2018 *Bioconjugate chemistry*.

**Figure 2 biomedicines-10-02055-f002:**
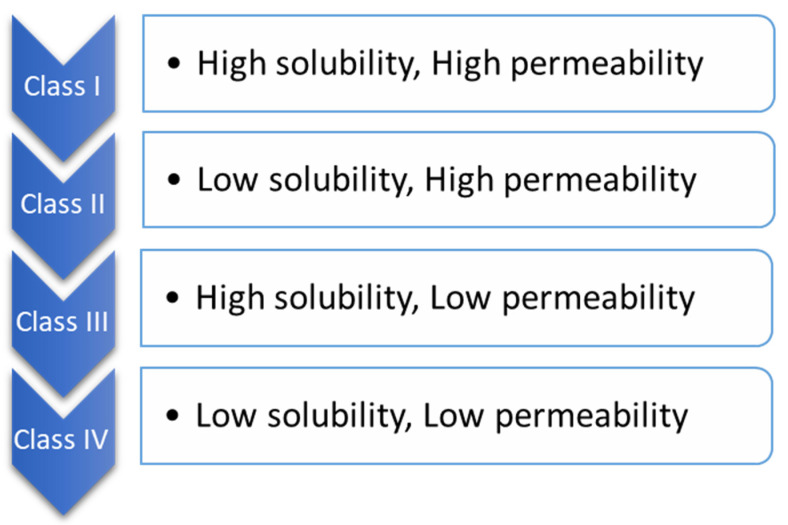
Biopharmaceutical classification.

**Figure 3 biomedicines-10-02055-f003:**
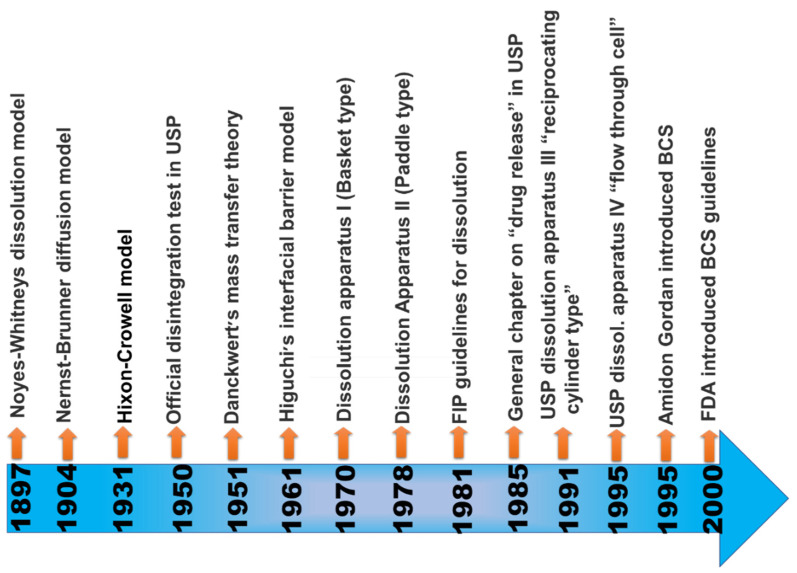
Evolutionary history of biopharmaceutical classification system [26].

**Figure 4 biomedicines-10-02055-f004:**
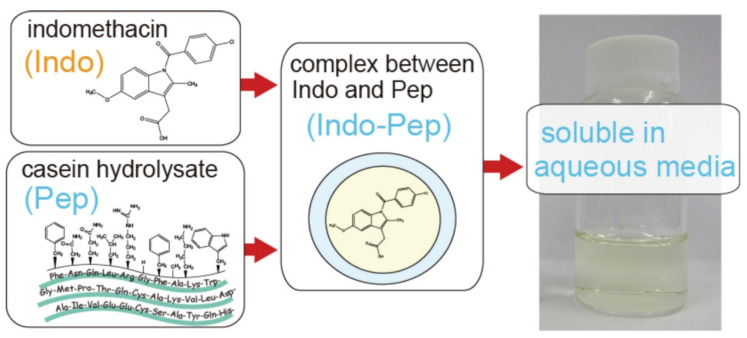
Enhancement of drug solubility of poorly aqueous soluble drugs through complexation with peptides, adapted with permission from [98].

**Figure 5 biomedicines-10-02055-f005:**
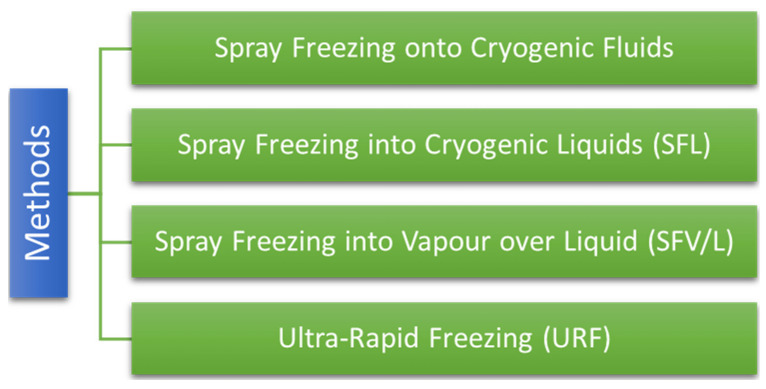
Various cryogenic methods.

**Table 1 biomedicines-10-02055-t001:** Examples of commercial amorphous drug-delivery products by FDA approval date.

Trade Name	Treatment	Drug (s)	Excipient (s)	Manufacturer (Year/Method)	References
ISOPTIN-SRE	Antihypertensive	Verapamil	HPC/HPMC	Abbott Laboratories, Chicago, IL, USA (1981/melt extrusion)	[2,16,17,18,19]
Cesamet	Anti-emetic, analgesic	Nabilone	PVP	Valeant Pharmaceuticals, Quebec, Canada (1985/melt extrusion)	[2,16,17,18,19]
Nivadil	Anti-hypertensive, major cerebral artery occlusion	Nilvadipine	HPMC	Fujisawa Pharmaceuticals Co., Ltd., Tokyo, Japan (1989/not available)	[2,16,17,18,19]
Sporanox	Antifungal	Itraconazole	HPMC	Janssen Pharmaceuticals Beerse, Belgium (1992/spray layering)	[2,16,17,18,19]
PROGRAF	Immunosuppressant	Tacrolimus	HPMC	Astellas Pharma Northbrook, IL, USA, Inc. (1994/spray drying)	[2,16,17,18,19]
REZULIN	Antidiabetic	Troglitazone	PVP	Pfizer/Parke-Davis, Detroid, MI, United States (1997/melt extrusion)	[2,16,17,19]
Afeditab	Anti-hypertensive	Nifedipine	PVP or poloxamer	Elan/Watson (2000/melt or absorb)	[17,19]
GRIS-PEG	Antifungal	Griseofulvin	PEG	Pedinol Pharmacal Inc., Novartis, Bridgewater, NJ, USA (2003/melt extrusion)	[2,16,17,19]
Nimotop	Anti-hypertensive	Nimodipine	PEG	Bayer Leverkusen, Germany (2006/spray drying)	[17]
KALETRA	HIV	Lopinavir, Ritonavir	PVP-VA	AbbVie Inc. Chicago, IL, USA (2007/melt extrusion)	[2,16,17,18,19]
Fenoglide	Anti-cholesterol	Fenofibrate	PEG	Veloxis Pharmaceuticals, Cary, NC, USA (2007/spray melt)	[17]
INETELENCE	HIV	Etravirine	HPMC	Tibotec, Janssen, Titusville, NJ, USA (2008/spray drying)	[2,16,17,18]
NORVIR	HIV	Ritonovir	PVP-VA	AbbVie Inc., Chicago, IL, USA (2010/melt extrusion)	[2,17,18,19]
ONMEL	Antifungal	Itraconazole	PVP-VA or HPMC	GlaxoSmithKline, Brentford, UK, Stiefel (2010/melt extrusion)	[17,18,19]
CERTICAN and ZORTRESS	Immunosuppressant	Everolimus	HPMC	Novartis Pharmaceuticals Basel, Switzerland (2010/spray drying)	[2,16,17,18,19]
INCIVEK	Antiviral; hepatitis C	Telaprevir	HPMCAS	Vertex Pharmaceuticals, Boston, MA, USA, Janssen (2011/spray drying)	[2,17,18,19]
ZELBORAF	Melanoma skin cancer	Vemurafenib	HPMCAS	Roche Basel, Switzerland (2011/co-precipitation)	[2,17,18,19]
KALYDECO	Cystic fibrosis	Ivacaftor	HPMCAS	Vertex Pharmaceuticals Boston, MA, USA(2012/spray drying)	[17,18,19]
NOXAFIL	antifungal	Posaconazole	HPMCAS	Merck, Kenilworth, NJ, USA (2013/melt extrusion)	[17,18]

Reprinted and adapted with permission from ref. [14]. Copyright 2018 *Bioconjugate chemistry*.

**Table 2 biomedicines-10-02055-t002:** Approaches for solubility enhancement.

Type		Advantages	Limitations	References
Crystal Engineering	Metastatic polymorph Co-crystal formation	Minimum amounts of surfactants and polymers are required for stabilization. High drug loading and high-energy systems are beneficial in drug dissolution.	Challenges in drug/polymer miscibility and excipient compatibility for a chosen drug. Physical instability upon storage.	[32,33]
Chemical Modification	Pro-drug formation	Improved drug solubility, lipophilicity, transported-mediated absorption. Has the potential to achieve site-specific delivery.	Limitations in producing screening and development. Associated with a high possibility for the formation of degradation by-products and lack of chemical stability. Disruption of solid-state crystallinity and polymorphism.	[34,35]
	Salt formation	The most commonly applied technique to increase solubility and the preferred approach for the development of liquid formulation. Enhances the dissolution rate by increasing the apparent intrinsic solubility of the drug. Ease of synthesis and low cost of raw material.	Restricted to weakly basic or acidic drugs; inappropriate for neutral-digested substances. After oral delivery, the medication is transformed back into either its free acid or basic form. Limitations in salt screening and the selection of optimal salt forms.	[36,37]
Particle size reduction	Micronization and nanosized drugs, e.g., NanoCrystal, DissoCubes	Easy to scale up and time efficient. Reduced drug degradation because the drug is in the crystalline solid state. Feasibility of formulation of a drug under different pharmaceutical dosage forms.	Physicochemical-related stability issues such as aggregation or a change in the solid state of the drug. The excess use of excipients as stabilizers may change the drug’s bioavailability, and pharmacological activity. Bulking care is essential, particularly during handling and transport.	[2,38,39]
Amorphization	Solid dispersion	Provides extra stability and protection of the drug during formulation. Enhanced solubility and dissolution rate compared with traditional crystal habit modification; it also retards agglomeration/crystallization of drug molecules due to its molecular level dispersion and steric hindrance interactions within the polymeric matrices.	Drugs that are high-energy amorphous tend to recrystallize and change into low-energy crystalline forms. The miscibility between the selected drug and polymeric matrices is required. Limited stability is a known drawback.	[40,41,42]
Solvent Composition	pH adjustment	The simple and powerful strategy for solubility adjustment of ionizable drugs. The level of ionization of the drug candidates enables full solvation of the target medication dose. This method works equally well with drug salts or the corresponding free basic or acid medicines.	The long-term effect on the drug stability. The distortion on physiological pH. The precipitation tendencies and incompatibility upon dilution.	[43,44,45]
	Co- solvent	Provides the optimum solubility for nonpolar drugs by reducing solvent polarity. The presence of a co-solvent can provide additional solubilization for drug solutions where pH manipulation is insufficient.	The use of co-solvents is limited to relatively few solvents. The risk of precipitation upon dilution. It may alter the pH and strength of the buffers that are contained in a drug formulation.	
Drug carrier systems	Micelles	Its hydrophobic core acts as a reservoir for lipophilic drugs. Ease of chemical modification and can be stimuli responsive.	The disintegration of micelles due to their dilution after oral administration, in vivo instability below the critical micelle concentration. Low drug loading.	[2,46,47]
	Nanoparticles	Increased solubility of lipophilic drugs, enhanced drug stability, sustained drug delivery, shielding of the drug cargo from enzymatic activity, prolonged retention in the gastrointestinal tract, improved mucoadhesiveness, overcoming multidrug resistance, the potential for targeting specific cells and uptake via M cells.	Challenges in biocompatibility and safety of polymeric carriers. Toxicity is a result of the high tissue accumulation of non-biodegradable NPs. Difficulties in optimizing the process parameters and scaling up the production into a pharmaceutical product.	[48,49,50]
	Cyclodextrins	Generally recognized as a safe (GRAS) excipient. Suitable for the generation of supersaturated drug solutions. Enhance both the physical and chemical stability of drugs and their shelf-life.	The requirement for a large amount of cyclodextrin compared to the drug to solubilize the drug. The weak binding and dissociation of complexes upon dilution in the GIT. The intact drug/cd complexes are unable to permeate the lipophilic epithelium membranes, which may result in low bioavailability, especially for BCS class III drugs.	[51,52,53]
	Lipid-based formulations (SLN, liposomes, SEDDS)	Non-immunogenic, biocompatible, can stimulate the secretion of bile salts, phospholipids and cholesterol, which form vesicles and micelles that then facilitate drug absorption, scalable and easily manufacturable.	Poor stability and short shelf life.	[54,55,56]

Reprinted and adapted with permission from ref. [57]. Copyright 2021, *Frontiers in Pharmacology*.

**Table 3 biomedicines-10-02055-t003:** Major techniques for solubility enhancement.

Major Solubility Enhancement		
Physical Modification	Chemical Modification	Miscellaneous Method
Particle size reduction Nanosuspension Modification of the crystal habit Drug dispersion by surfactant Complexation Cryogenic technique	pH adjustment Hydrotrophy Co-crystallization Co-solvency Salt formation Novel excipients	Super critical fluid technology Micellar solubilization Direct capsule filling Electrospinning Dropping method solution

**Table 4 biomedicines-10-02055-t004:** Marketed products based on nanoparticle techniques [69].

Drug	Indications	Inventor Company	Drug Delivery Company	Trade Name
Methyl phenidate HCl	CNS stimulant	Novartis Basel, Switzerland	Elan Nanosystems	Ritalin^®^
Morphine sulfate	Psychostimulant drug	King Pharmaceuticals Bristol, UK	Elan Nanosystems	Avinza^®^
Aprepitant	Anti-emetic	Merk & Co. Kenilworth, NJ, USA	Elan Nanosystems	Emend^®^
Tizanidine HCl	Muscle relaxant	Acorda New York, NY, USA	Elan Nanosystems	Zanaflex Capsules^®^
Megestrol	Anti-anorexic	Par Pharmaceutical New York, NY, USA	Elan Nanosystems	Triglide^®^
Fenofibrate	Hypercholesterolemia	ScielePharma Inc., Atlanta, GA, USA	IDD-P Skyepharma	Trilide^®^
Dexmethylphenidate HCl	Attention deficit hyperactivity disorder (ADHD)	Novartis Basel, Switzerland	Elan Nanosystems	Focalin^®^
Fenofibrate	Hypercholesterolemia	Abbott Laboratories Chicago, IL, USA	Abbott Laboratories	Tricor^®^
Rapamycin, sirolimus	Immunosuppressant	Wyeth Madison, NJ, USA	Elan Nanosystems	Rapamune^®^

**Table 5 biomedicines-10-02055-t005:** Approaches for the enhancement of drug solubility and oral bioavailability by nanotechnology [69].

Company	Formulation Approach Based on Nanotechnology	Description
American Biosciences (Blauvelt, NY, USA)	Nanoparticle albumin-bound technology	Paclitaxel albumin nanoparticles
BioSante Pharmaceuticals (Lincolnshire, IL, USA)	For the enhancement of oral bioavailability of hormones/proteins and vaccines, nanoparticles of calcium phosphate were developed	Calcium phosphate nanoparticle
Baxter Pharmaceuticals (Deerfield, IL, USA)	Nanoedge technology: particle size reduction by homogenization, micro-precipitation, lipid emulsion and other dispersed systems.	Nano-lipid emulsion
Imbedding (Burlingame, CA, USA)	Silicon membranes were used for implantable drug delivery Membrane pore size (10–100 nm)	Stretchable silicon nanomembrane

**Table 6 biomedicines-10-02055-t006:** Manufacturing technologies of solid dispersions.

Technology	Drug Molecule	BCS Class	Trade Name	Formulation	Therapeutic Use
Nanocrystal (wet media milling)	Rapamycin	II	Rapamune	Tablets	Immunosuppressant
	Aprepitant	IV	Emend	Capsules	Antiemetic
	Finofibrate	II	Tricor	Tablets	Antilipidemic
	Megestrol acetate	II	Megace ES	Oral suspension	Hormonal Therapy
High-pressure homogenization	Fenofibrate	II	Triglide	Tablets	Antilipidemic
Melt extrusion	Verapamil HCL	I	Isoptin SRE	Tablets	Antihypertensive
	Nifedipine	II	Adalat SL	Capsules	Antihypertensive
	Troglitazone	II	Rezulin	Tablets	Antilipidemic
Melt Adsorption	Nifedipine	II	Afeditab	Tablets	Antihypertensive
Melt granulation	Fenofibrate	II	Fenoglide	Tablets	Antilipidemic
	Tacrolimus	II	LCP- Tacro	Tablets	Immunosuppressant
Spray drying	Intellence	IV	Etravirine	Tablets	Antiviral
	Itraconazole	II	Sporanox	Capsules	Antifungal
	Nilvadipine	II	Nivadil	Capsules	Antihypertensive
	Tacrolimus	II	Prograf	Capsules	Immunosuppressant
Lyophilization	Olanzapine	II	Zyprexa	Tablets	Antipsychotic
	Ondansetron	II	Zofran ODT	Tablets	Antiemetic
	Piroxicam	II	Proxalyoc	Tablets	Anti-inflammatory

Reprinted and adapted with permission from ref. [57]. Copyright 2021, *Frontiers in Pharmacology*.

**Table 7 biomedicines-10-02055-t007:** Marketed parenteral microemulsion products.

Drug	Therapeutic Area	Product Name	Company
Cyclosporine A	Immunomodulation	Restasis	Allergan (Dublin, Ireland)
Prostaglandin-E1	Vasodilator	Liple	Green Cross (Tokyo, Japan)
Diazepam	Sedation	Diazemuls	Braun Melsungen (Melsungen, Germany)
Propofol	Anesthesia	Propofol Diprivan	Baxter (Illinois, United States)
Dexamethasone Palmitate	Corticosteroid	Limethason	Green Cross (Tokyo, Japan)
Perflurodecalinþ Perflurotripropylamine	Analgesia	Fluosol-DA	Green Cross (Tokyo, Japan)
Etomidate	Anesthesia	Etomidat	Dumex (Lillerød, Denmark)
Vitamins A, D, E and K	Nutrition	Vitalipid	Kabi (Bad Homburg, Germany)
Flurbiprofen	Analgesia	Lipfen	Green Cross (Tokyo, Japan)

**Table 8 biomedicines-10-02055-t008:** Examples of various drugs developed by SLN technology.

Drug	Lipid Utilized	Biopharmaceutical Application
5-Fluoro uracil	Dynasan 114 and Dynasan 118	Prolonged release in simulated colonic media
Ibuprofen	Stearic acid, triluarin and tripalmitin	Stable formulation with low toxicity
Apomorphine	Glycerylmonostearate, polyethylene glycol monostearate	Enhanced bioavailability in rats
Idarubicin	Emulsifying wax	Delivery of oral proteins
Calcitonin	Trimyristin	Improvement of the efficacy of proteins
Lopinavir	Campritol 888 ATO	Bioavailability enhanced
Clozapine	Trimyristin, tristearin and tripalmitin	Improvement of bioavailability
Nimesulide	Glycerylbehanate, glyceryltristearate, palmitostearate	Sustained release of the drug
Cyclosporin A	Glycerylmonostearate and glycerylpalmitostearate	Controlled release
Progesterone	Monostearin, oleic acid and stearic acid	Potential for oral drug delivery
Gonadotropin release hormone	Monostearin	Prolonged release
Repaglinide	Glycerylmonostearate and tristearin	Reduced toxicity

**Table 9 biomedicines-10-02055-t009:** Lipid excipients are frequently used in lipid-based nanocarriers.

Excipient	Chemical	Type of Carrier	Comments	References
Soybean oil	Triglycerides (long-chain)	Nanoemulsions	Liquid, good biocompatibility, minimal physiological impact, weak solubilizing capacity	[125,126,127]
Olive oil	Triglycerides (long-chain)	Nanoemulsions	Liquid, healthy, high monounsaturated fatty acids, and simple to emulsify	[126,128,129,130,131]
Hemp oil	Medium/long-chain triglycerides blended with low-molecular weight lipids	Nanoemulsions	The liquid contains tocopherols, tocotrienols, phyrosterols, phospholipids, and other important fatty acids, good hydrophilicity, and self-emulsifiability.	[132,133]
Caprylic/capric triglycerides	Triglycerides (medium-chain)	Nanoemulsions	Liquid, solubilizing capacity, compatible with other lipids, easy to emulsify.	[134,135,136,137,138,139]
Captex^®^ Series	Triglycerides (medium-chain)	Nanoemulsions	Liquid, fine solubilizing and emulsifying capacities, miscible with other lipids	[140,141,142]
Capmul MCM	Mono/diglycerides (medium-chain)	Nanoemulsions	Liquid, an excellent solvent powder for many organic compounds, can use as an emulsifier.	[143,144,145,146]
Capmul MCM C8	Glycerol monocaprylate	Nanoemulsions	Liquids, property similar to that of Capmul MCM.	[147,148,149]
Maisine^TM^ 35-1	Glycerol monolinoleate	SEDDS	Liquid, solubilizer, bioavailability enhancer, oil phase in SEDDS	[150,151,152,153]
Peceol^TM^	Glyceryl monolete	SEDDS; NLCs; Cubosomes	Liquid, lipid dispersion agent, oil-soluble surfactant, moisturizer	[154,155,156]
Lauroglycol^®^ 90	Propylene glycol monolaurate	Nanoemulsions; SEDDS; NLCs	Liquid, water-insoluble surfactant of SEDDS, solubilizer, bioavailability enhancer, skin penetration solubilizer enhancer.	([157,158,159]
Capryol^TM^ series	Propylene glycol monocaprylate	Nanoemulsions; SEDDS; NLCs	Liquid, properties similar to that of Lauraglycol^®^ 90	[160,161,162]
Labrafil M 1944 CS	Oleoyl polyoxyl-6 glycerides	Nanoemulsions; SEDDS; NLCs	Liquid, water dispersible surfactant, able to self-emulsify good miscibility with other lipids, bioavailability enhancer, solubilizer, co-emulsifier.	[163,164,165]
Lecithin	Phosphatidylcholine blended with a small amount of other lipid components	Liposomes, phytosomes, lipid nanoparticles	Semi-solid, an amphiphilic lipid, used as a vesicle-forming material, solubilizing, emulsifying and stabilizing agents.	[166,167,168,169,170]
Gelacire^®^ series	Lipid blends consisting of mono-, di-, or triglycerides and fatty acid macrogolglycerides	SEDDS, SLNs, NLCs	Semi-solid, non-ionic water soluble surfactant for solid/semi-solid dispersions and SEDDS, bioavailability enhancer, micelle-forming material, solubilizing and wetting agent	[160,171,172]
Monostearin	Glyceryl monostearate	SLNs, NLCs	Solid, lipid matrix for SLNs and NLCs; thickening, solidifying and control release adjusting agent.	[149,173]
Precirol^®^ ATO 5	Glyceryl distearate	SLNs; NLCs	Solid, lipid matrix for SLNs and NLCs, hydrophobicity and melting point greater than glyceryl monostearate.	[174,175]
Compritol^®^ 888 ATO	Glyceryl behenate	SLNs, NLCs solid lipid dispersions	Solid, high-melting point lipid, used for the preparation of SLNs and NLCs, lipid matrix for sustained release, used as atomized powders.	[176,177,178]
Trilaurin	Glyceryl trilaurate	SLNs, NLCs	Solid, lipid matrix for SLNs and NLCs, sustained release material, thickening agent.	[179,180,181]
Cetyl palmitate	Palmityl palmitate	SLNs, NLCs	A solid, wax-like substance, used for preparation of SLNs and NLCs	[182,183]
Tripalmitin	Glyceryl tripalmitate	SLNs; NLCs	Solid, lipid matrix of SLNs and NLCs, skin-conducting agent.	[184,185]

Reprinted and adapted with permission from ref. [186]. Copyright 2018, *Pharmaceutics*.

**Table 10 biomedicines-10-02055-t010:** List of nanocarrier applications in oral drug delivery.

Nano-System	Composition	Drug Molecule Size (nm)	Size (nm)	Cell Line/Animal Model	Disease or Targeted Organ	References
Dendrimers	G3.5 PAMAM	SN38	-	Caco-2 cells and HT-29/female CD-1 mice	Colorectal cancer metastases	[196]
	Ethylene diamine and methyl acrylate	SN38 camptothecin	13	CD-1 mice	Oral chemotherapy of hepatic colorectal cancer metastases	[197]
	PAMAM	Short hairpin RNA	107–315	Tca8113 cells/BALB/c nude mice	Oral cancer therapy	[198]
Micelles	Polyethylene oxide-polypropylene oxide-polyethylene oxide (PEO-PPO-PEO)	Paclitaxel	180	Female C57BL/6J mice	Oral cancer therapy	[199]
	*N*-octyl-*O*-sulfate chitosan (NOSC)	Paclitaxel		Caco-2/SD rats	Improved oral bioavailability	[200]
	Bovine-casein	Celecoxib, Paclitaxel	20	Human N-87 gastric cancer cells	Rheumatoid arthritis, osteoarthritis, and gastric carcinoma	[201,202]
	Tocopherol succinate glycol chitosan conjugates	Ketoconazole	101	Caco-2 cell monolayer	Improved oral bioavailability	[203]
Mixed micelles	Pluronic copolymers and LHR conjugate	Paclitaxel	140	MCF-7 cells	Oral anticancer delivery system	[204]
Vesicles	PLA-P85-PLA	Insulin	178	OVCAR-3 cells/diabetic mice	Oral insulin delivery	[205]
Liposomes	Lecithins	Curcumin	263	Sprague-Dawley (SD) rats	Improved oral bioavailability	[206]
SLN	Iyceryl monostearate (GMS)	Vinpocetine	70–200	Male Wistar rats	Improved oral bioavailability	[207]
Polymeric microspheres	Chitosan and alginate	Insulin	5–7 µm	Male SD rats	Diabetes mellitus	[208]
Polymeric nanoparticles	PLGA	Cyclosporine	143 nm	Male SD rats	Improved oral bioavailability	[209]
	Silica	Resveratol	90 nm	Caco-2 cell monolayer	Enhanced the solubility, permeability and anti- inflammatory activity of resveratrol encapsulated in NPs	[210]
Multifunctional polymeric nanoparticles	Galactose-modified trimethyl chitosan-cysteine conjugates with various galactose grafting densities	shRNA and siRNA	130–160 nm	Caco-2 cells/tumour-bearing mice	Targeted treatment of hepatoma	[211]
	Mannose-modified trimethyl chitosan-cysteine (MTC) conjugates	Tumor necrosis factor-α (TNF-α) siRNA	152.9 nm	Caco-2 cells, RAW 264.7 (monocyte/macrophage-like cells)/acute hepatic injury-induced mice	Treatment of systemic inflammatory conditions	[212]
	Lectin-conjugated PLGA-NPs	Betamethasone	475 nm	TNBS- induced colitis mice	Treatments of ulcerative colitis and inflammatory bowel disease	[213]

Reprinted and adapted with permission from ref. [57]. Copyright 2021, *Frontiers in Pharmacology*.

**Table 11 biomedicines-10-02055-t011:** List of patented formulations related to nanoparticles for oral drug delivery.

Patent Number	Assignee	Invention	References
WO2008073558A2	Johns Hopkins University, USA	The invention provided new orally bioavailable smart NPs for the delivery of poorly soluble drugs, showing improved pharmacokinetics and bioavailability.	[212]
WO2015067751A1	NanoSphere Health Sciences Inc., USA	Investigation disclosed the composition and development method for nutraceuticals encapsulated with phospholipid-based NPs by the emulsification method.	[214]
US20120003306A1	NancMega Medical Co., USA	The report disclosed a protein/peptide delivery system composed of chitosan and poly-y-glutamic acid (y-PGA). The NPs were suggested to enhance the epithelial permeability and thus are efficient for oral drug delivery.	[215]
WO2004098564A2	University of Illinois, USA	Reported the development of biodegradable NPs containing streptomycin with high loading efficiency of 50% or higher for tuberculosis treatment. The NPs can also contain other aminoglycoside drugs, which are a known substrate for the multidrug efflux mediated by P-glycoprotein (Pgp).	[216]
US7674767B2	Samyang Biopharmaceuticals Co., Korea	The invention described the composition and preparation of orally administrable NPs containing complexes of water-soluble drugs and counter-ion substances. The NPs enhanced drug entrapping and resistance against lipases, thereby increasing drug absorption.	[217]
WO2015023797A9	Northwestern University, USA	The patent disclosed the development and evaluation of drug-loaded nanostructures comprising an inorganic core and a lipid layer shell. The NPs showed potential in the treatment of cancer, vascular diseases and infectious diseases.	[218]
WO2014197640A1	South Dakota State University, USA	Disclosed the composition and preparation method of core-shell NPs. These NPs comprise food-grade proteins along with therapeutic agents suitable for pedatrics.	[219]
WO2007042572A1	Advanced In Vitro Cell Technologies S.A., Spain	The insertion described NPs comprising chitosan and heparin prepared by ionic gelation method. The NPs were stable in gastrointestinal fluids and presented excellent in vivo effectiveness and bioavailability.	[220]
CN102120781B	China Pharmaceutical University, China	The invention related to the preparation of oral insulin NPs. The NPs mainly contained N-amino acid chitosan as a carrier and insulin for the treatment of diabetes. The NPs were stable after oral administration with a better effect of reducing blood sugar in vivo.	[221]
US10420731B1	King Saud University, Saudi Arabia	The invention described the synthesis and preparation method of lignin NPs cross-linked and stabilized by citric acid for oral administration. The NPs improved the oral bioavailability of curcumin by increasing curcumin solubility and stability, sustaining its release, enhancing intestinal permeability, and inhibiting Pgp-mediated efflux.	[222]
WO2011034394A2	JW Pharmaceuticals Co., Korea	The invention reported the preparation of oxaliplatin-loaded NPs using supercritical fluid gas technology for oral chemotherapy.	[223]
WO2010015688A1	BioAlliance Pharma Co., USA	The patent disclosed the composition and preparation method of a chemotherapeutic formulation containing polymer and cyclic oligosaccharide capable of complexing and delivering anticancer drugs for effective cancer treatments.	[224]

Reprinted and adapted with permission from ref. [57]. Copyright 2021, *Frontiers in Pharmacology*.

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
