# Peer review of "Bioavailability Enhancement Techniques for Poorly Aqueous Soluble Drugs and Therapeutics"

_biomedicines, 2022, doi:10.3390/biomedicines10092055_

Round 1
Reviewer 1 Report
The Manuscript ID: biomedicines-1867275 discussed abut the several means of improvement of solubility/bioavailability of BCS II/IV drugs. The overall information is good for readers. Before acceptance, I would like to notify the authors regarding spelling mistakes as follows.
1. Table 1, 8-10 not mentioned in the discussion.
2. Line 144: "which are differs" need to correct
3. Line 138: "is subject to " need to correct
4. Table 2: Minimal amounts or amount?
5. Line 243-244: Nanosuspension is suitable for hydrophilic drugs?
6. Line 308: N.C.E.s with higher "availability" or bioavailability?
7. Line 490: perfect adoptation of "host" of guest molecule?
8. Line 599: Kneeing or kneading?
9. Table 7: Nimesulide?
10. Table 10, Ref 218: oxalplatin or Oxaliplatin??
11. Some references DOI is avilable, some do not have.
Author Response
Reviewer 1
Comment
The Manuscript ID: biomedicines-1867275 discussed several means of improving solubility/bioavailability of BCS II/IV drugs. The overall information is good for readers. Before acceptance, I would like to notify the authors regarding spelling mistakes as follows.
Response: Thank you for highlighting the spelling mistakes, now we have corrected all the spelling mistakes.
Comment 1. Table 1, 8-10 not mentioned in the discussion.
Response: Now, we have mentioned the table 1, 8-10 in text.
Comment 2. Line 144: "which are differs" need to correct
Response: Now, we have corrected the text.
Comment 3. Line 138: "is subject to " need to correct
Response: Now, we have corrected the text.
Comment 4. Table 2: Minimal amounts or amount?
Response: Now, we have corrected the text.
Comment 5. Line 243-244: Nanosuspension is suitable for hydrophilic drugs?
Response: Now, we have corrected the text. And write the text as “Nanosuspension is suitable for hydrophobic drugs”
Comment 6. Line 308: N.C.E.s with higher "availability" or bioavailability?
Response: Now, we have corrected the text. And the sentence is written as “N.C.E.s with a higher bioavailability”
Comment 7. Line 490: perfect adaptation of "host" of the guest molecule?
Response: Now, we have corrected the text.
Comment 8. Line 599: Kneeing or kneading?
Response: Now, we have corrected the text.
Comment 9. Table 7: Nimesulide?
Response: Now, we have corrected the text.
Comment 10. Table 10, Ref 218: oxaliplatin or Oxaliplatin??
Response: Now, we have corrected the text.
Comment 11. Some references DOI is available, and some do not.
Response: Now we have included all the DOI which is available, some references do not have DOI available (in the case of books and patents).
Reviewer 2 Report
The research article entitled “Bioavailability Enhancement Techniques for Poorly Aqueous Soluble Drugs and Therapeutics” provides information on various drug delivery systems with their advantages. The article also highlights drug solubility, bioavailability, The Biopharmaceutics Classification System (BCS), etc. The manuscript lacks a few important topics which need to be addressed such as biomolecules-based delivery systems like proteins, and hydrogels, the Importance of targeted drug delivery systems in improving Bioavailability with minimizing toxic side effects, etc. The article has many grammatical and sentence errors, and the language organization needs to be improved. For these reasons, I conclude that the paper should undergo minor revision
1. Please make sure that all keywords have been used in the abstract and the title.
2. The authors are required to provide a section about targeted drug delivery systems and their importance for improving Bioavailability with minimizing toxic side effects.
3. Authors can add a section on protein-based nanoparticles in drug delivery applications. Important information like FDP-approved Abraxane, https://doi.org/10.1016/j.molliq.2021.116623, https://doi.org/10.1016/j.foodchem.2018.11.124
https://doi.org/10.3390/pharmaceutics12070604
4. The reference format in the manuscript was found not as per the journal format. need to be corrected. In References 5, 208-219, the volume and page number are missing. Still many missing. Careful and vigorous verification is required.
5. Please improve the conclusion with clear Future perspectives and strategies with more emphasis on improving Bioavailability.
6. There are many grammatical and sentence errors in the article, and the language organization needs to be improved.
Author Response
Comment
The research article entitled “Bioavailability Enhancement Techniques for Poorly Aqueous Soluble Drugs and Therapeutics” provides information on various drug delivery systems with their advantages. The article also highlights drug solubility, bioavailability, The Biopharmaceutics Classification System (BCS), etc. The manuscript lacks a few important topics which need to be addressed such as biomolecules-based delivery systems like proteins, and hydrogels, the Importance of targeted drug delivery systems in improving Bioavailability with minimizing toxic side effects, etc. The article has many grammatical and sentence errors, and the language organization needs to be improved. For these reasons, I conclude that the paper should undergo minor revision.
Response:
Comment 1. Please make sure that all keywords have been used in the abstract and the title.
Response: Now we have modified the keywords and all the keywords have been used in the abstract and title.
Comment 2. The authors are required to provide a section about targeted drug delivery systems and their importance for improving Bioavailability with minimizing toxic side effects.
Response: We are preparing a separate review on “targeted drug delivery systems and their importance for improving Bioavailability with minimizing toxic side effects.”
Comment 3. Authors can add a section on protein-based nanoparticles in drug delivery applications. Important information like FDP-approved Abraxane, https://doi.org/10.1016/j.molliq.2021.116623, https://doi.org/10.1016/j.foodchem.2018.11.124
https://doi.org/10.3390/pharmaceutics12070604
Response: Now we have added a section on protein-based nanoparticles in drug delivery applications. Please see section 2.1.5.3.
Comment 4. The reference format in the manuscript was found not as per the journal format. need to be corrected. In References 5, 208-219, the volume and page number are missing. Still many missing. Careful and vigorous verification is required.
Response: We have now corrected the references as per journal format.
Comment 5. Please improve the conclusion with clear Future perspectives and strategies with more emphasis on improving Bioavailability.
Response: Now, we have done corrections in the conclusion.
Comment 6. There are many grammatical and sentence errors in the article, and the language organization needs to be improved.
Response: Now we have corrected the grammatical, and sentence errors and improved the language organization in the article.
This manuscript is a resubmission of an earlier submission. The following is a list of the peer review reports and author responses from that submission.